# Diversity in Accessions of *Panicum miliaceum* L. Based on Agro-Morphological, Antioxidative, and Genetic Traits

**DOI:** 10.3390/molecules24061012

**Published:** 2019-03-13

**Authors:** Bimal Kumar Ghimire, Chang Yeon Yu, Seung Hyun Kim, Ill-Min Chung

**Affiliations:** 1Department of Applied Life Science, Konkuk University, Seoul 143-701, Korea; bimal_g12@yahoo.com (B.K.G.); kshkim@konkuk.ac.kr (S.H.K.); 2Bioherb Research Institute, Kangwon National University, Chuncheon 200-701, Korea; cyyu@kangwon.ac.kr

**Keywords:** *Panicum miliaceum*, morphological characteristics, antioxidant activities, phenolic content, total flavonoid content, genetic diversity

## Abstract

The genetic diversity and antioxidant potential of *Panicum miliaceum* L. accessions collected from different geo-ecological regions of South Korea were evaluated and compared. Antioxidant potential of seeds was estimated by the 1,1-diphenyl-2-picrylhydrazyl (DPPH) and 2,2′-azino-bis-3-ethylbenzthiazoline-6-sulfonic acid (ABTS) radical scavenging assays and total phenolic content was determined by the Folin–Ciocalteu method. Total phenolic content (TPC) in 80% methanolic extracts ranged from 16.24 ± 0.86 to 58.04 ± 1.00 mg gallic acid equivalent (GAE)/g of the sample extracts and total flavonoid content (TFC) varied from 7.19 ± 1.05 to 52.56 ± 1.50 mg quercetin equivalents (QE) mg/g of the sample extracts. DPPH radical scavenging capacity of the extracts from the 15 accessions of *P. miliaceum* varied from 206.44 ± 7.72 to 2490.24 ± 4.641 mg GAE/g of the sample extracts and ABTS radical scavenging capacity ranged from 624.85 ± 13.1 to 1087. 77 ± 9.58 mg GAE/g of the sample extracts. A wide range of genetic variation was observed as measured by Shannon’s information index (I), number of effective alleles (Ne), number of observed alleles (Na), expected heterozygosity (He), unbiased expected heterozygosity (uHe). The observed variation in the bioactive properties, morphological traits, and genetic diversity among the accessions may provide useful information for breeding programs seeking to improve bioactive properties of *P. miliaceum*.

## 1. Introduction

*Panicum miliaceum* L., also known as proso millet, belongs to the *Graminacea* family and is an ancient crop that has been cultivated for more than 7000 years [1]. It is also considered an economically important crop that is cultivated mainly in China, Eastern Europe, India, Russia, and Northern America [2]. This plant is mainly used as functional food owing to its high protein content [3,4]. In addition, it is also used as a raw material for liquor and beer, animal fodder, and as a sizing agent in the textile industry in Africa and Asia [5,6]. It is a gluten-free food whose mild flavor is used in food industries of America and Europe [7]. Moreover, millet requires relatively low amounts of water and nutrients for growth and can be cultivated in a wide range of altitudes [6]. It is considered a rich source of micronutrients such as iron (Fe), zinc (Zn), copper (Cu), manganese (Mn), vitamins, and trace elements [8]. Therefore, this plant is receiving increasing attention from food industries. Moreover, previous studies on this plant reported about its anticancer and antidiabetic properties and its ability to prevent coronary heart disease and liver diseases [9,10,11].

The rapid loss of genetic diversity of plant species due to changing environment creates an urgent need to conserve existing important alleles and valuable phenotypic traits of genetic resources and to gather information about the genetic variation of different accessions. Studies on genetic diversity not only increase our knowledge but are also vital for improving yield, disease resistance, and nutritional value [12], as they facilitate the genetic integration of new useful genes [13], and are important for future breeding programs [14]. Moreover, characterizing genetic variation of plant species helps select superior germplasm for the rational use of genetic resources [15].

Characterizing morphological traits is very important when assessing the agronomic value and taxonomically classifying of plant species [16]. Morphological traits are also useful tools in studying geographical patterns in large gene bank collections of plant species [17,18]. However, morphological characterization has its own disadvantages, as it is time-consuming and varies according to the environmental changes and low level of polymorphism, low heritability, late expression, limited discriminative power, and lower potential to measure relatedness and genetic similarity [19]. DNA fingerprinting markers play a major role in revealing polymorphisms. The selection of accessions is more accurate when genetic markers are used rather than morphological traits, as genetic markers can be obtained from all tissues of organisms, allow early detection, and can produce highly specific pattern of bands for each individual. Genetic markers, also called DNA markers, show variation among individuals based on variation in DNA sequences [20,21,22]. There are many kinds of molecular markers available, including amplified fragment length polymorphism (AFLP), restriction fragment length polymorphism (RFLP), random amplified polymorphic DNA (RAPD), simple sequence repeats (SSR), single nucleotide polymorphisms (SNP), inter-simple sequence repeats (ISSR), and diversity arrays technology (DArT) markers, among which ISSR markers are most reliable as they are simple, cost-effective, and highly polymorphic, and can augment plant breeding programs [23].

ISSR markers have become reliable tools in the field of DNA fingerprinting and analyzing genetic diversity of plants. Since these markers do not require sequence information, variations may be found at several loci simultaneously, are microsatellite-sequence specific, are reliable for DNA profiling, especially for closely related species [24], and are highly reproducible and polymorphic, ISSRs are used extensively in taxonomic studies [25] to determine phylogenetic relationships and plant breeding programs [26,27,28]. In contrast, the use of RAPD represents a very cost-effective, rapid, and efficient approach in detect higher levels of polymorphisms and unique alleles in plants [20]. The usefulness of these molecular markers has been extensively demonstrated for detecting genetic variations in several plant species [29,30,31,32]. Applying two marker systems gives more reliable and correct information about genetic diversity. Moreover, the omissions recorded by certain markers can be solved or minimized by using more than one marker [33].

Previous studies on the genetic diversity in *P. miliaceum* were employed by using genetic markers [34,35,36,37]. Even though genetic diversity was assessed using different types of DNA markers, the genetic diversity of *P. miliaceum* has been inadequately assessed. Variation in the chlorophyll content in the plant can be used as an indicator of plant vigor [38]. These pigments are also associated with the antioxidant properties of plants [38]. A number of studies have reported that phytochemicals, such as phenolic compounds, flavonoids, and isoflavones not only act as antioxidant compounds [39], but are also safe for consumption as dietary supplements. Other studies have reported natural antioxidants extracted from plants that effectively inhibited or reduced the formation or scavenging of free radicals and carcinogens [40]. *P. miliaceum* has a wide geographical range, which results in variations in nutritional value, biological activity, and agro-morphological traits. To our knowledge, there is no report on the chlorophyll content, morphological variation, and antioxidant and genetic diversity assessed using a combined molecular marker technique in *P. miliaceum* accessions. Moreover, the antioxidant profiles of *P. miliaceum* accessions that originate from different parts of the Korean peninsula have not been fully examined in earlier studies. Previous research on *P. miliaceum* accessions were limited and incomplete in the absence of these parameters. Therefore, the objectives of the present study were to compare the bio-morphological and genetic traits in the accessions and to screen biological activities and their correlations with total phenolic content (TPC) and total flavonoid content (TFC). By screening the accessions, we can have diverse adapted germplasm for safe and productive *P. miliaceum* production. Furthermore, the present comparative study might identify accessions with useful genes for breeding programs, and may be able to introduce a commercial variety to farmers.

## 2. Results

### 2.1. Morphological Variation in P. miliaceum

A total of 18 quantitative and qualitative traits of the 15 accessions of *P. miliaceum* studies are shown in Table 1, Table 2 and Table 3. 13.33% of the accessions (PM-04 and PM-05) took a long period to begin blooming and attain full bloom, while 46.67% of the accessions showed early bloom start and full bloom (PM-01, PM-02, PM-06, PM-08, PM-14, PM-015). A significant positive correlation was observed between bloom begin and full bloom (*r* = 0.995, *p* < 0.01) (Table 3). In this study, days to bloom was moderately positively correlated with leaf length and plant height (*r* = 0.453 and *r* = 0.420, *p* < 0.01, respectively), indicating that leaf number of accessions was a critical factor that directly influenced the initiation of flowering. Plant height, an important parameter in plant breeding, showed a wide variation ranging from 111.03 ± 2.00 cm to 186.30 ± 3.50 cm. Among all the accessions, PM-07 displayed the tallest height (186.30 ± 3.50 cm), while accession PM-09 showed the shortest height (111.03 ± 2.00 cm). In this report, plant height was shown to be highly positively correlated with culm length (*r* = 0.980, *p* < 0.01), and moderately positively correlated with stem diameter (*r* = 0.621, *p* < 0.01), total number of leaves (*r* = 0.608, *p* < 0.01), average leaf length (*r* = 0.630, *p* < 0.01) indicating that these parameters were critical factors that directly influenced the growth of *P. miliaceum* accessions.

The accessions had a wide range of variation in culm length, ranging from 74. 00 ± 2.00 cm (in PM-09) to 148.50 ± 1.50 cm (in PM-07). Culm length showed significant moderate positive correlations with stem diameter, number of leaves, and leaf length (*r* = 0.595 and *r* = 0.626, *r* = 0.652, *p* < 0.01, respectively). However, culm length weakly correlation with the bloom begin and full bloom (*r* = 0.190 and *r* = 0.157 *p* < 0.01), although there was no significant correlation between these parameters at *p* < 0.01. In the present study, spike length and width were also distinctive morphological markers to distinguish among the studied *P. miliaceum* accessions. The spike length ranged from 34.8 ± 2.80 cm to 52.5 ± 7.50 cm, while spike width ranged from 2.6 ± 0.60 cm to 5.0 ± 1.00 cm. The highest spike length and width was observed in PM-12 (52.5 ± 7.5 cm and 5.0 ± 1.0 cm, respectively) (Figure 1). Other quantitative traits such as spike type and tiller number were not significantly different among the accessions. The majority of accessions exhibited a single tiller. Leaf color and leaf orientation were the only qualitative parameters and a significant polymorphism was observed within the accessions. Leaf orientation was almost horizontal with respect to the stem axis in the majority of accessions. Accessions PM-02 and PM-15 were remarkably different from the rest of the accessions with leaves slightly below the horizontal axis, pointing downwards. A wide range of variation in the leaf length and width was observed among the 15 accessions. Accession PM-10 displayed the longest leaf length (54.17 ± 2.00 cm), with higher leaf length/width ratio (18.48 ± 0.07), while PM-14 displayed the shortest leaf length (31.00 ± 2.00). The average number of leaves and average stem diameter also varied significantly (*p* < 0.01) among the 15 accessions. The average stem diameter of accessions ranged from 0.4 ± 0.0 cm to 1.41 ± 0.3 cm in PM-08 and PM-13, respectively. Leaf length was shown to be highly positively correlated with leaf width and stem diameter (*r* = 0.763 and *r* = 0.764, *p* < 0.01, respectively).

The accessions exhibited a wide range of variation in seed color. The majority of accessions exhibited creamy seed coat. The average weight of 1000 seeds per accessions varied from 4.41 ± 0.1 g in PM-03 to 6.36 ± 0.06 g in PM-08, however, 1000 seed weight exhibited moderate negative correlations with bloom begin and full bloom both (*r* = −0.482 and −0.477, respectively), although there was no significance between these parameters at *p* < 0.01 (Table 3). 

### 2.2. Variations in Chlorophyll a, Chlorophyl b, and Carotenoid Content

Relative chlorophyll pigment (Chl a, Chl b, ratio of Chl a/chl b, and carotenoids) contents were measured in the 15 accessions (Table 4). In general, a wide range of variability was observed among the accessions. Results revealed a higher concentration of Chl a and Chl b in PM-02 and PM-09, while PM-11 exhibited the lowest amount of Chl a and Chl b accumulation. However, no significant differentiation was observed in carotenoid concentration within the studied accessions. A significant and high positive association was observed between Chl a and Chl b with carotenoids (*r* = 0.997, *p* < 0.01). A strong negative interrelationship was also observed between Chl a and Chl b with carotenoid contents in the accessions (*r* = −0.912 and *r* = −0.935, *p* < 0.01, respectively).

### 2.3. Principal Component Analysis

In this study, PCA was applied to assess the morphological variations using 18 morphological traits obtained from the fifteen accessions of *P. miliaceum* (Figure 2). The first and second principal components scores were 35.540% and 18.506% of the total variance, respectively. Along axis 1 of the PCA analysis, three accessions formed a group (PM-02, PM-04, PM-07) for the positive side of the PC1 axis, mainly characterized by the morphological traits such as higher culm length, leaf length and width, and average number of leaves, indicating that these accessions were closely related genotypes. Five accessions (P-03, PM-05, PM-09, PM-10, PM-15) formed a separate group on the positive side of the PC2 axis, mostly sharing the characteristics of longer time to bloom start and full bloom. Six other accessions (PM-01, PM-05, PM-08, PM-11, PM-13, PM-14) formed another group on the negative side of the PC2 axis, mainly characterized by morphological traits such as higher weight of 1000 seeds. Accession PM-12, a sole representative on the negative side of the PC1 axis, strongly contributed to morphological traits, such as higher spike length, plant height, stem diameter, and spike length. The present study revealed that the studied accessions did not group according to their geographical origins.

### 2.4. Screening of Total Phenolic Content, Total Flavonoid Content and Antioxidant Potential of P. miliaceum Accessions

Total phenolic contents of 15 *P. miliaceum* accessions are presented in the Table 5. Total phenolic content in 80% methanolic extracts at a concentration of 1000 ppm ranged from 16.24 ± 0.86 to 58.04 ± 1.00 mg GAE per 1 g of sample. PM-03 had a significantly higher total phenolic content (58.04 ± 1.00 mg GAE per 1 g of sample). A wide range of significant variations (*p* < 0.001) in TFC were observed among the studied accessions, ranging from 7.19 ± 1.05 to 52.56 ± 1.50 mg quercetin equivalent per 1 g sample. 

At the accessions level, PM-05 displayed the highest TFC (52.56 ± 1.50 mg quercetin equivalent per 1 g of sample) and the lowest amount of total flavonoid contents (7.19 ± 1.05 mg quercetin equivalent per 1 g sample) was observed in PM-04. DPPH radical scavenging capacity of the extracts from the 15 accessions of *P. miliaceum* are shown in the Table 5. The RC_50_ values among the studied accessions ranged from 206.44 ± 7.72 to 2490.24 ± 4.641 µg mL^−1^. The extracts from accession PM-05 exhibited the highest antioxidant capacity as represented by a lower RC_50_ value (206.44 ± 7.72 µg mL^−1^), while the lowest DPPH radical scavenging activity was observed in the extracts of PM-04 (2490.24 ± 4.64 µg mL^−1^). The ABTS radical scavenging capacity of the different accessions of *P. miliaceum* varied considerably (Table 5). The highest ABTS scavenging capacity was observed in PM-14 as represented by lower RC_50_ values (624.85 ± 13.1 µg mL^−1^), while the lowest ABTS scavenging capacity (1087.77 ± 9.58 µg mL^−1^) was recorded in extracts of PM-04. In the present study, it is likely that collecting accessions from different geo-ecological regions with different altitudes may have significantly influenced these parameters.

### 2.5. Correlation between Antioxidant Capacity, Total Phenolic, and Total Flavonoid Contents

Pearson’s correlation coefficients were obtained to determine the relationship between antioxidant capacity, total phenolic content, and total flavonoid contents (Table 6). A significant, high correlation (*p* < 0.01) was observed between DPPH scavenging activity with total phenolic and total flavonoid contents (*r* = 0.732 and *r* = 0.933, *p* < 0.01, respectively). In contrast, a weak and nonsignificant association was observed between ABTS scavenging activity with TPC and TFC (*r* = 0.370 and *r* = 0.268, *p* < 0.01, respectively).

#### 2.5.1. Genetic Variability Details from ISSR Markers

A total of 29 ISSR primers were screened for the ISSR analysis and 22 of them resulted in amplification of DNA from accessions. A wide variation was recorded in the amplification products (200 to 800 bp). A total of 160 bands were generated, 30% of which showed polymorphism. All the generated bands were distinct and varied in size and type (monomorphic and polymorphic). The highest number of scorable bands (15) was observed in ISSR-809 primers (Figure 3), while the least number of bands were scored from ISS-854. Out of the 22 scorable primers used, 22.22% showed 100% polymorphism (Table 7). A wide range of variation was also observed from determining Shannon’s information index (I), number of effective alleles (Ne), number of observed alleles (Na), expected heterozygosity (He), and unbiased expected heterozygosity (uHe) (Table 9). The average Shannon’s information index value (I) was 0.581 ± 0.006 (ranged from 0.337 to 0.693). The average number of effective alleles (Ne) was 1.675 ± 0.015 (ranged from 1.233 to 1.998). The observed average number of different alleles (Na) was 2.00. The average expected heterozygosity (He) and unbiased expected heterozygosity (uHe) in the accessions were 0.395 ± 0.006 and 0.408 ± 0.006, respectively.

#### 2.5.2. Genetic Variability Details from RAPD Markers

Eleven RAPD primers were selected for amplifying DNA of *P. miliaceum* accessions (Table 8). A total of 46 bands were generated, 27.27% of which showed polymorphism. The number of polymorphic bands varied widely among the primers used. The maximum number of polymorphic bands (seven bands per primer) was observed in RAPD-1 (Figure 4), while the least number of polymorphic bands (3 bands per primer) was generated using RAPD-7. The size of polymorphic bands varied from 250 to 900 bp. A wide range of variation was also observed in Shannon’s information index (I), number of effective alleles (Ne), number of observed alleles (Na), expected heterozygosity (He), and unbiased expected heterozygosity (uHe) (Table 9). The average Shannon’s information index value (I) was 0.578 ± 0.29 (ranged from 1.383 to 1.999. The average number of effective alleles (Ne) was 1.662 ± 0.048 (ranged from 1.383 to 1.99). The observed average number of different alleles (Na) was 2.00 ± 0.00. The average expected heterozygosity (He) and unbiased expected heterozygosity (uHe) in the accessions was 0.391 ± 0.018 and 0.396 ± 0.01, respectively.

#### 2.5.3. Genetic Variability Details from RAPD + ISSR Combined Data

In the present study, we used two different marker systems (ISSR and RAPD), to maximize the efficiency of detecting polymorphisms and to define genetic similarity within the accessions. A total of 206 bands were generated by combining both ISSR and RAPD primers. None of the primers generated identical band patterns, indicating a wide range of diversity within the accessions. The observed differences in loci number by using RAPD and ISSR marker indicate differences in the genetic sequence in the studied accessions. The average Shannon’s information index value (I) was 0.601 ± 0.007 (ranged from 0.538 to 0.644). The average number of effective alleles (Ne) was 1.700 ± 0.018 (ranged from 1.546 to 1.823). The observed average number of different alleles (Na) was 2.00 ± 0.00. The average expected heterozygosity (He) and unbiased expected heterozygosity (uHe) in the accessions was 0.411 ± 0.006 and 0.412 ± 0.006, respectively (Table 10). In the present study, combining molecular marker systems produced higher informative classification than single markers alone.

#### 2.5.4. UPGMA Cluster Analysis Using ISSR and RAPD Marker

Based on the ISSR marker data, similar coefficient values among the accessions ranged from 0.78 to 0.95 (Figure 5). These data were used to construct a dendrogram using the UPGMA method. All accessions can be classified into five major groups. 

Group I consisted of five accessions (PM-01, PM-02, PM-04, PM-07, PM-08). Group II comprises five accessions (PM-09, PM-10, PM-11, PM-12, PM-13). Group III consisted of a single accession (PM-06), characterized by strong antioxidant properties. Group IV comprises two accessions (PM-14, PM-15) that mostly shared characteristics of similar seed color, tiller number, color of leaves, average number of leaves, bloom start, and full bloom. Group V consisted of two accessions (PM-03, PM-05), the majority of which were characterized by identical stem diameter, plant height, full bloom duration, spike length and width, and leaf orientation.

Based on RAPD markers, similar coefficient values among the accessions ranged from 0.18 to 0.84. From the data, five distinct groups were observed (Figure 6). Group I consisted of six accessions (PM-01, PM-15, PM-07, PM-08, PM-09, PM-11), the majority of which were characterized by a short duration until bloom start and full bloom. Group II was comprised of three accessions (PM-02, PM-03, PM-05), mostly sharing the characteristic of short spike width. Group III consisted of two accessions (PM-06, PM-10), sharing the common feature of identical leaf width, while group IV and V comprised a single accession in each group. The polymorphisms shown by RAPD analysis among the accessions indicate that this tool is effective in distinguishing *P. miliaceum* accessions and determining their genetic diversity.

#### 2.5.5. RAPD and ISSR Combined Data for Cluster Analysis

Based on the combination of ISSR + RAPD markers, similar coefficients among the fifteen accessions ranged from 0.75 to 0.92 (Figure 7). These values were used to construct a dendrogram using UPGMA. All the accessions could be clustered into five main groups. Group I consisted of four accessions (PM-01, PM-04, PM-07, PM-08), while group II comprises five accessions (PM-09, PM-10, PM-11, PM-13, and PM-12), which were distinguished from other groups because of identical bloom begin and full bloom. Group III consisted of a single accession (PM-06), while group IV consisted of two accessions (PM-14 and PM-15), which were distinguished from other accessions of other groups due to identical short duration of bloom begin and full bloom. Group V consisted of three accessions (PM-02, PM-03, PM-05) that mostly shared the characteristic of short spike width. In this study, the total number of clusters and identification of accessions in each cluster by using morphological traits and molecular markers varied. The distant accessions such as PM-01 and PM-15 identified on the basis of combined ISSR and RAPD markers were found to be identical in terms of bloom start, full bloom, and culm length, whereas genetically identical accessions (PM-14 and PM-15) differed in terms of plant height, leaf length, leaf width, ration of leaf length/width, culm length, stem diameter, and spike length and width. The present study observed no correlation between the agro-morphological characteristics and genetic diversity.

## 3. Discussions

### 3.1. Morphological Variation in P. miliaceum

Characterizing agro-morphological traits is not only helpful in breeding programs but is also useful for improving the biological and functional properties of food [41]. The characteristics and diversity of fifteen *P. miliaceum* accessions were assessed using agro-morphological traits. In our study, there was a wide range of variation in the morphological traits among the accessions. Full bloom period was significantly different (*p* < 0.05) but varied within the studied accessions. These results were consistent with findings of Vetriventhana and Upadhyaya [6]. It has been argued that the full bloom period is important because it provides information about the time of flowering, pollination, and seed development and dispersal [42]. In this study, days to bloom was moderate, positively correlated with leaf length and plant height (*r* = 0.453, *p* < 0.01), indicating that leaf length of accessions was a critical factor that directly influenced the initiation of flowering. A significant positive correlations have been found between plant height and bloom begin (*r* = 0.420, *p* < 0.01). It has been reported that short height and early bloom of crops are generally associated with higher ability to compete with nutrition and light for higher yield, require a small cultivation area, and have increased density compared to taller accessions [43,44]. Other quantitative traits such as culm length were significantly different among the accessions and showed significant moderate positive correlations with stem diameter, number of leaves, and leaf length. In the present study, spike length and width were also distinctive morphological markers to distinguish among the studied *P. miliaceum* accessions. Leaf color and leaf orientation were the only qualitative parameters and a significant polymorphism was observed within the accessions. Leaf orientation was almost horizontal with respect to the stem axis in the majority of accessions. It has been suggested that leaf orientation is directly associated with plant growth and yield [45,46]. Moreover, our study corroborated the findings of Trivedi et al. [47] and Lagler et al. [48], who reported phenotypic variation including leaf length, leaf width, plant height, bloom begin, 1000 seed weight, and seed color in the *P. miliaceum* accessions. However, in this study, 1000 seed weight exhibited moderate negative correlations with bloom begin and full bloom both. In a similar study, a negative correlation was observed between 1000 grain weight and days to maturity [49] and plant height of wheat accessions [50]. Other studies found that variations in agro-morphological traits such as plant height leaf orientation and duration of maturity were clearly associated with survival and yield of *P. miliaceum* plants [51,52,53]. According to these studies, variation in these parameters helps accessions of various altitude and origin to adapt to different agro-climatic conditions. Moreover, a number of previous studies attributed the variation in morphological traits to genetic, developmental, and environmental factors [54,55].

### 3.2. Variations in Chlorophyll a, Chlorophyl b, and Carotenoid Content

Information about the chlorophyll contents of the plant accessions can be used as an indicator of plant vigor and contribute significantly in improving the plant biomass and grain yield [38]. It has been reported that the amount of these pigments are associated with genetical and environmental factor [56] and related to the photosystem II and oxidative stress of the plant [57,58]. In the present study, the accessions exhibited a wide range of variation in relative chlorophyll pigment (Chl a, Chl b, and ratio of Chl a/chl b). However, No clear trend was observed in the carotenoid content of the accessions. Our study corroborated the findings of Trivedi et al. [47], wherein they reported wide variation in these parameters among *P. miliaceum* accessions. In the present studies, a moderate and significantly positive correlation between chlorophyll contents and weight of 1000 seed (data not shown), indicating that these pigments are associated with the grain yield of *P. miliaceum* accessions. Previous studies have shown that a positive correlation between chlorophyll content and photosynthesis rate [59]. Beside role in photosynthesis, chlorophylls and carotenoids are known for its antioxidant properties [60]. In the present studies, we observed a weak positive correlation between chlorophyll contents and antioxidant activity, (though not significant), indicating that these metabolites contributed partially in the antioxidant activity of *P. miliaceum* accessions. Moreover, other researchers have concluded that greater chlorophyll a and b accumulation increases light absorption and ROS removal from plants, ultimately facilitating plant growth and increased the grain yield [61,62].

### 3.3. Screening of Total Phenolic Content, Total Flavonoid Content and Antioxidant Potential of P. miliaceum Accessions

Previous studies reported the close association between TPC and antioxidant potential, which is significantly influenced by variation in the plant of origin, its climatic conditions, and environment and plant populations [63,64]. Meanwhile, other studies emphasized the importance of growth season, altitudinal variation, and duration of plant maturity for variations in antioxidant activity [65,66,67]. In the present study, it is likely that collecting accessions from different geo-ecological regions with different altitudes may have significantly influenced these parameters. Moreover, our study corroborated the findings Chandrasekara and Shahidi [68], wherein they reported a wide variation in antioxidant activity in *P. miliaceum* collected from different geographical locations. Furthermore, they observed a higher, positive correlation between DPPH and ABTS (*r* = 0.820, *p* < 0.01), which is good agreement with the observations of the present study. Several studies reported a linear correlation between the antioxidant capacity of a plant with its types and concentration of phenolic compounds [69,70,71]. In the present study, the wide range of variation observed in the antioxidant capacity of the accessions can be associated with variations in the total phenolic and total flavonoid contents within them. It has been reported that the number of OH groups, arrangement of OH groups, and availability of electron-donating and withdrawing substituents in the phenolic compounds significantly determines the antioxidant capacity of plant extracts [72,73,74]. Aromatic rings and hydroxyl groups vary widely in the phenolic compounds, thus causing differences in the antioxidant capacity of plants [75,76]. Other studies pointed out that differences in genotypic and environmental factors during growth account for variation in the antioxidant potential in accessions [69,70,71]. Thus, present data can serve as an initial material for plant breeding programs of *P. miliaceum* for higher antioxidant activity and chlorophyll contents.

### 3.4. Genetic Variability Details from ISSR and RAPD Markers

In order to understand the distribution and genetic variation, an adequate knowledge about the existing plant population and traits is essential for the improvement of genetic resources [77]. Study of the plant population diversity considering only morphological traits is inadequate and inconvincing due to environmental influences [78,79]. Moreover, DNA fingerprinting methods employed in the study of genetic variation is a most steady and reliable methods across the plant accessions [80]. Use of ISSR and RAPD markers have become reliable tools in the field of DNA finger printing and analyzing genetic diversity, germplasm conservation and genomic mapping [81,82].

In the present study, a total of 22 ISSR primers and eleven RAPD primers were selected for amplifying DNA of *P. miliaceum* accessions. Both ISSR and RAPD markers showed more than 70% polymorphism. The results indicate that both ISSR and RAPD markers were effective in identifying the genetic variability in the *P. miliaceum* accessions. The average Shannon’s information index value (I), average number of effective alleles (Ne), average expected heterozygosity (He) and unbiased expected heterozygosity (uHe) detected by ISSR primers was higher than that of the RAPD primers. Based on comparison with the earlier reports, the present study essentially did not agree with the results of Trivedi et al. [47], where they reported low number of effective alleles (Ne) and low Shannon’s information index (I) values using ISSR marker in the studied *P. miliaceum* accessions, which might be due to different genetic backgrounds and better primer selection for the genetic diversity assessments in this study. The result indicates that ISSR markers are more effective for assessing genetic diversity. In terms of number of bands generated per primer, ISSR primers were a more efficient marker system than RAPD primers for determining polymorphic bands and to determine genetic diversity, which is corroborated with reports in peanut [83], chickpea [84], barley [85], pepper [86], and rice [87]. It has been reported that ISSR markers produce more polymorphism by amplifying microsatellite-rich regions of DNA due to mutations during replication [88]. On the other hand, comparatively, RAPD is less responsible and causes variable DNA band patterns, and is also highly influenced by experimental conditions [89,90]. In contrast, using RAPD markers, M’Ribu et al. [91] reported great polymorphism among proso millet cultivars. Similar to the present study, de Wet et al. [92] and Reddy et al. [93] reported high genetic diversity and morphological variation in proso millet accessions using RAPD primers. In the present study, combining molecular marker systems produced higher informative classification than single markers alone. The high level of polymorphism and genetic variation values using both ISSR and RAPD markers were in agreement with report for *Stevia rebaudiana* [33], *Cucumis sativus* [94], *Cucurbitaceae* [95], *Pleurotus eryngii* [31], and *Vigna unguiculata* [96]. It has been suggested that applying diverse marker types can provide more accurate results, thus helping evaluate genetic diversity efficiently [33]. It is worth emphasizing here that the combined used of ISSR and RAPD markers on the accessions of *P. miliaceum* and comparing the efficiency in determining genetic diversity had not been performed previously in *P. miliaceum*.

### 3.5. ISSR and RAPD Combined Data for Cluster Analysis

The dendogram clusters obtained could show the genetic relationship between the 15 accessions of *P. miliaceum.* The present study reveals that all 15 accessions were clustered into five major groups in both ISSR and RAPD methods. In a similar study, Trivedi et al. [47] reported different cluster patterns in accessions of *P. miliaceum* using ISSR markers. In other studies, Karam et al. [34] reported different cluster patterns in accessions of *P. miliaceum* using AFLP markers, whereas Liu et al. [35] reported four major groups at a genetic similarity level of 0.633 using SSR markers. In this study, the total number of clusters and identification of accessions in each cluster by using morphological traits and molecular markers varied. The distant accessions such as PM-01 and PM-15 identified on the basis of combined ISSR and RAPD markers were found to be identical in terms of bloom start, full bloom, and culm length, whereas genetically identical accessions (PM-14 and PM-15) differed in terms of plant height, leaf length, leaf width, ration of leaf length/width, culm length, stem diameter, and spike length and width. The lack of correlation between these two different approaches agreed with the report of Pathak et al. [97] in lychee cultivars, where they observed different identification results using molecular markers and morphological traits. Meanwhile, others argued that variability among accessions may be caused by uncontrolled spread of plant materials and longtime sexual reproduction and different degrees of domestication causing regular expression of certain genes and inactivation of other genes [98].

### 3.6. Conclusions

The present study is the first to assess the relationships among antioxidant capacity, morphological characteristics, and genetics of accessions of *P. miliaceum* using RAPD and ISSR markers. Wide differences in loci number by using RAPD and ISSR indicated differences in the genetic sequences of the studied accessions. This important information about the variations in the antioxidant capacity, morphological, and genetic diversity may be used in germplasm conservation and to assist molecular breeding programs for *P. miliaceum* with improved bioactive capacity. However, in the present study, a relatively small number of accessions was used for assessing biological activity, morphological characteristics, and genetic characterization of *P. miliaceum* accessions. Therefore, our future work will focus on considering a greater number of accessions for elucidating the relationship between these parameters.

## 4. Material and Methods

### 4.1. Chemicals, Standard Compounds and Solvents

All commercial standard compounds with at least 99% purity used for analyzing individual phenolic compounds were purchased from Sigma Aldrich Chemical Co. (St. Louis, MO, USA) and Extrasynthese (Genay Cedex, France). HPLC-grade methanol hexane, ethyl acetate, and butanol were supplied from Avantor–J. T. Baker (Phillipsburg, NJ, USA). Water used in this study was of analytical grade purified by a Milli-Q Water Purification System (Millipore, Bedford, MA, USA). Other chemical compounds such as DPPH and ABTS, which were used for assessing biological activity, were purchased from Avantor–J. T. Baker (Phillipsburg, NJ, USA).

### 4.2. Plant Materials

In this study, 15 accessions of *P. miliaceum* collected from different eco-geographical regions of Korea were grown from 2014–2017 at the Agriculture Research Field of Kangwon National University, South Korea (Table 10). All the accessions used in this study were grown naturally under ideal, similar plant growth conditions using the same field management practices at 99 m elevation.

### 4.3. Evaluation of Morphological Traits in P. miliaceum Accessions

Prior to cultivating the accessions, experimental units were prepared in a completely randomized block design, each unit consisting of 10 replicates. The length of each row was maintained at 70 m with 1 m between adjacent rows with approximately 80-cm gaps between seedlings. The rainfall recorded during the cultivated period was 200 nm with the average minimum and maximum temperature of 20 °C and 35 °C, respectively. The soil of the experimental field was maintained at pH of 6.1 and irrigated regularly (once a week) using a drip-irrigation system. To increase the nutrient contents of soil, the recommended dose of fertilizers (N:P:K = 15%:15%:15%) at a rate of 125 kg ha^−1^ were supplied when the land was prepared. To control weeds, landscape fabric was used to cover the spaces between the rows. The weeds that appeared within the rows were hand-picked at regular intervals. The recommended dose of pesticides was used to control pests. Quantitative and qualitative characteristics of field-grown accessions were recorded during the second week of October in 2014–2017. The morphological traits of each accession were recorded from five randomly selected plants of each accession. The different quantitative parameters considered for the study included: fresh weight, dry weight, plant height, leaf length, leaf width, number of leaves, leaf color, leaf orientation, ratio of leaf length to leaf width, culm length, number of nodes, tiller number, weight of 1000 seeds, full bloom time, bloom beginning, spike width, and spike length. Qualitative traits such as seed shape and seed color of each accession were rated (visually) and analyzed. The average height of each accession was measured using a ruler from the base of the plant to the apex of the stem. The emergence of flower (bloom beginning) was measured as the time duration (days) between the sowing and the emergence of the first flower. Full bloom was the time duration (days) between planting and emergence of approximately 50% of the flowers from each accession. The average weight of 1000 seeds was measured from 10 randomly selected samples from each *P. miliaceum* accession.

### 4.4. Screening of Chlorophyll Contents

To extract chlorophyll a, b, and carotenoid, 0.5 g of leaf sample was taken, and homogenized with 10 mL of 80% acetone. The homogenized sample was centrifuged for 4000 rpm for 10 min at 4 °C. The supernatants were collected in the cuvette. The chlorophyll a, b, and carotenoid content was spectrophotometrically analysed using a Shimadju UV-1800 UV-VS spectrophotometer (Shimadzu, Kyoto, Japan) by following method described previously [99]. The concentration of chlorophyll-a, chlorophyll-b, and carotenoids was calculated using following equation:Chlorophyll-a = 12.25A_663.2_ − 279A_646.8_Chlorophyll-b = 21.5A_646.8_ − 5.1A_663.2_(1)
Carotenoids = (1000A_470_ − 1.82Chl_a_ − 85.02Chl_b_)/198(2)

### 4.5. Screening of Total Phenolic Concentration

The TPC of different extracts was determined using Folin–Ciocalteu’s phenol reagent by following the method described previously [100]. Briefly, 100 µL of plant extracts (1000 ppm) or standard compounds (gallic acid) were mixed with 200 µL of Folin–Ciocalteu’s phenol reagent and allowed stand for 5 min at room temperature (25 °C). After five minutes, 300 µL of Na_2_CO_3_ (20% *w*/*v*) solution was added to the mixture to stop the reaction. After incubating for 40 min at room temperature, the absorbance value was recorded at 765 nm against a blank solution (200 µL of 80% methanol). Gallic acid at various concentrations (1, 5, 10, 100 mg/L) was used for obtaining a calibration curve. Total phenolic contents of samples were expressed as mg gallic acid equivalent (GAE)/g of sample. All experiments were performed in triplicate.

### 4.6. Screening of Total Flavonoids Concentration

The TFC of different extracts was determined using a spectrophotometric method [101]. Briefly, 500 mL of plant extract (1000 ppm) or standard (quercetin) were added to 300 µL of AlCl_3_ and 200 µL of potassium cyanide. The mixtures were incubated for 30 min and absorbance value of the solution was measured immediately at 510 nm against a blank. The blank consisted of 500 mL of 80% methanol. A calibration curve was drawn using various concentrations of quercetin (1, 3, 5, 10 mg/L) and the TFCs of tested samples were expressed as mg quercetin equivalent (Qu)/G sample. All experiments were performed in triplicate.

### 4.7. Screening of Antioxidant Activity of P. miliaceum Accessions

#### 4.7.1. Evaluation of DPPH Assay

Antioxidant activity of fifteen *P. miliaceum* accessions was determined using a 1,1-diphenyl-2-picryl-hydrazyl radical (DPPH) assay by following the methods described previously [102]. Briefly, 200 µL of plant crude extracts at different concentrations (ranging from 10,000 ppm to 100 ppm) or standard compounds mixed with 300 µL of 0.004% freshly prepared methanol solution of DPPH in a 96-well plate. The mixture was incubated at room temperature (25 °C) for 30 min in the dark. The absorbance value of each test solution was measured at 515 nm using a V530 UV-VIS spectrophotometer (Jasco, Tokyo, Japan) connected with an automated plate reader. α-tocopherol was used as a positive control. The DPPH scavenging potential of each sample was measured using the following equation:RC_50_ = Ab_control_ − Ab_sample_/Ab_control_(3)
where Ab_control_ is the absorbance value of the control reaction (only DPPH radical solution) and Ab_sample_ is the absorbance value of the plant sample (plant extract mixed with DPPH radical solution).

#### 4.7.2. Evaluation of ABTS Assay

An ABTS assay was performed using a previously described previously [103] with some modifications. Initially, a working solution of ABTS (7.4 mM) potassium persulfate solution (2.6 mM) was mixed in an equal ratio (1:1, *v*/*v*) and allowed to react for 12 h at room temperature (25 °C) in the dark. The mixture was then diluted in methanol and the absorbance at 734 nm of the solution was adjusted to 0.600 ± 0.01. Then, 2.5-mL aliquots of extracts were added to 2.5 mL of ABTS solution. After gentle agitation, the solution was allowed to stand for 2 h in the dark. The absorbance of the mixture was measured at 734 nm. A standard curve was generated using a Trolox standard solution at various concentrations (500–1000 µM). The radical scavenging activity of each sample was expressed as mmol Trolox equivalent/100 g. All the experiments were performed in triplicate. The ABTS capacity of each sample was calculated as follows:RC_50_ = Ab_control_ − Ab_sample_/Ab_control_(4)
where Ab_control_ is the absorbance of the control reaction (ABTS radical solution with no sample and positive control) and Ab_sample_ is the absorbance of the test compound (ABTS radical solution with a sample or positive control).

### 4.8. DNA Extraction

Young leaves each from fifteen *P. miliaceum* accessions were collected for DNA extraction. DNA of individual accessions was extracted by following the standard CTAB method [104]. Approximately one gram of fresh leaf from 15 accessions was crushed individually in liquid nitrogen and suspended with extraction buffer (100 mM Tris-HCl, pH 8.5, 1.4 mM NaCl, 20 mM EDTA, 2% CTAB, and 0.2% β-mercaptoethanol) and incubated in a water bath for 45 min at 65 °C. After incubation, the supernatant was transferred to another 1.5 mL Eppendorf tube and an equal volume of ice-cold chloroform–isoamyl alcohol was added and the mixture were inverted and then spun at 11,952× *g* for 10 min. The supernatant was placed into another Eppendorf tube and an equal volume of ice-cold isopropanol was added to the mixture and incubated at −20 °C for 40 min. Then, the tube was centrifuged at 17,226× *g* for 10 min. After discarding the supernatant, 70% ice-cold ethanol was added to the DNA pellet. The DNA pellet was air dried and dissolved in 30 µL of nuclease-free water. The quality and concentration of DNA genomic DNA of all accessions was determined using a UV-VIS spectrophotometer (Jasco V530 UV-VIS) and 0.8% agarose gel electrophoresis. Extracted DNA was diluted to 5 ng/µL using 1 mmol/L TE buffer.

#### PCR Amplification and Electrophoresis

A total of 29 ISSR primers used to amplify the DNA of 15 accessions of *P. miliaceum* were obtained from the Bioherb Research Institute, Kangwon National University, South Korea. A total of 15 highly polymorphic ISSR markers were selected for this study. PCR amplification for the genomic DNA of each accession was performed in a 20-µL reaction volume containing 50 ng genomic DNA, 2 µL 1X PCR buffer, 1 U Taq DNA polymerase; 1 µM of primer, and 300 µM of dNTPs. PCR conditions for DNA amplification were as follows: initial denaturation step for 4 min at 95 °C, followed by 45 cycles of denaturation for 30 s at 94 °C, primer annealing for 45 s at 48–52 °C, and extension for 2 min at 72 °C, followed by a final extension for 10 min at 72 °C. The amplified products were loaded onto the 1% gel in 0.5X TBE buffer. The amplified PCR products were electrophoretically separated on 0.8% agarose in 0.5X TBE buffer for 20 min at 25 V and observed under UV light.

A total of eleven RAPD primers used to amplify the DNA of 15 accessions of *P. miliaceum* were obtained from Bioherb Research Institute (Kangwon National University, Chuncheon, South Korea). RAPD amplification reactions were performed in a 20-µL reaction volume containing 50 ng genomic DNA, 2 µL 1X PCR buffer, 1 U Taq DNA polymerase, 1 µM of primer, and 300 µM of dNTPs obtained from Bioneer (Daejeon, South Korea). The amplification conditions were an initial denaturation at 94 °C for 5 min; 45 cycles of 30 s at 94 °C, 1 min at 42 °C, and 2 min at 72 °C; a final extension of 5 min at 72 ° C, followed by storage at 4 °C.

### 4.9. Statistical Analysis

Each experiment was performed in triplicate. The data obtained from the experiments were expressed as mean ± standard deviation. Quantitative data were statistically analyzed using one-way analysis of variance (ANOVA). Significant differences between the obtained data were determined by using Duncan’s multiple range test at *p* < 0.05 (SPSS ver. 20.0, SPSS Inc., Chicago, IL, USA). Correlations among the morphological traits, antioxidant activities, total phenolic content, and total flavonoid content were obtained by calculating Pearson’s correlation coefficient using SPSS software ver. 20.0. Principal component analysis (PCA) of quantitative morphological traits was performed using SPSS software ver. 20.0. Cluster analysis using the unweighted pair group method with arithmetic averages (UPGMA) was performed using SPSS software ver. 20. Shannon’s information index (I), number of effective alleles (Ne), number of observed alleles (Na), expected heterozygosity (He), and unbiased expected heterozygosity (uHe) were determined using GenAlEx software. NTSYS software v. 2.00 [105] (Exeter Publishing, Ltd., Setauket, NY, USA) was used to generate a UPGMA dendrogram.

## Figures and Tables

**Figure 1 molecules-24-01012-f001:**
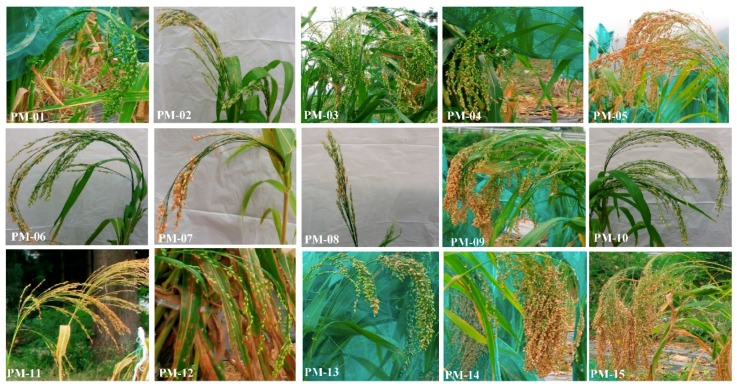
Variation in spike morphology in the 15 accessions of *P. miliaceum* studied.

**Figure 2 molecules-24-01012-f002:**
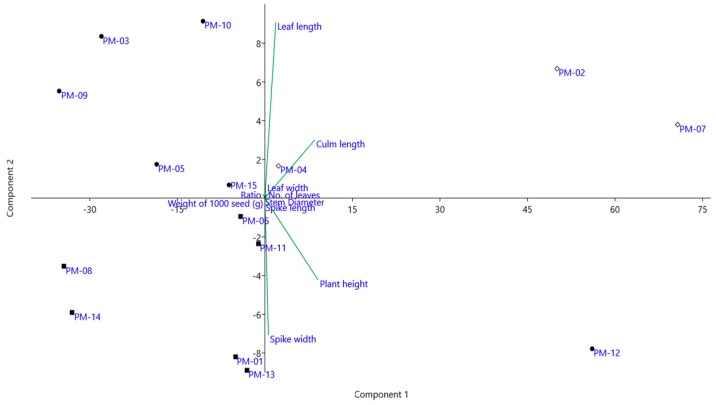
Principal component analysis in the 15 accessions in *P. miliaceum* studied based on morphological characteristics.

**Figure 3 molecules-24-01012-f003:**
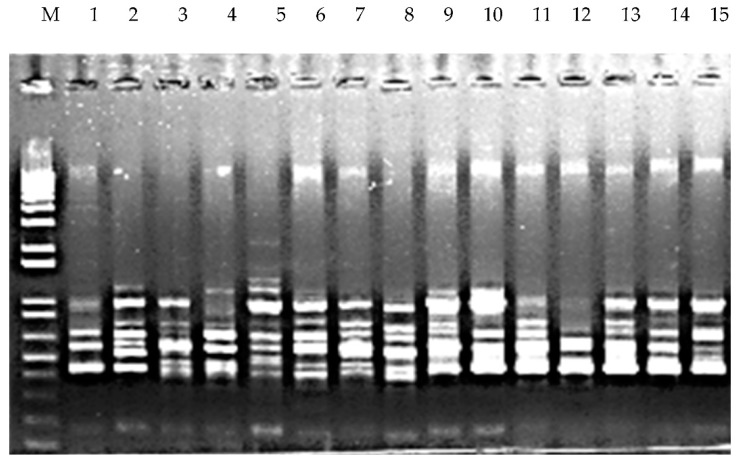
Banding pattern generated by the ISSR-844 primer. M: DNA ladder, 1–15 represent the DNA banding pattern of fifteen accessions of *P. miliaceum.*

**Figure 4 molecules-24-01012-f004:**
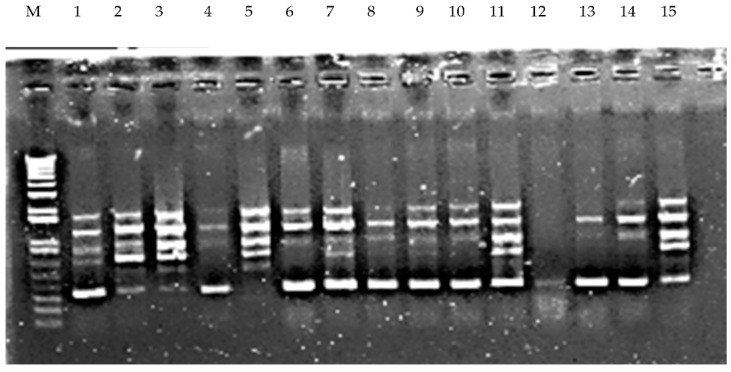
Banding pattern generated by the RAPD-1 primer. M: DNA ladder, 1-15 represent the DNA banding pattern of fifteen accessions of *P. miliaceum.*

**Figure 5 molecules-24-01012-f005:**
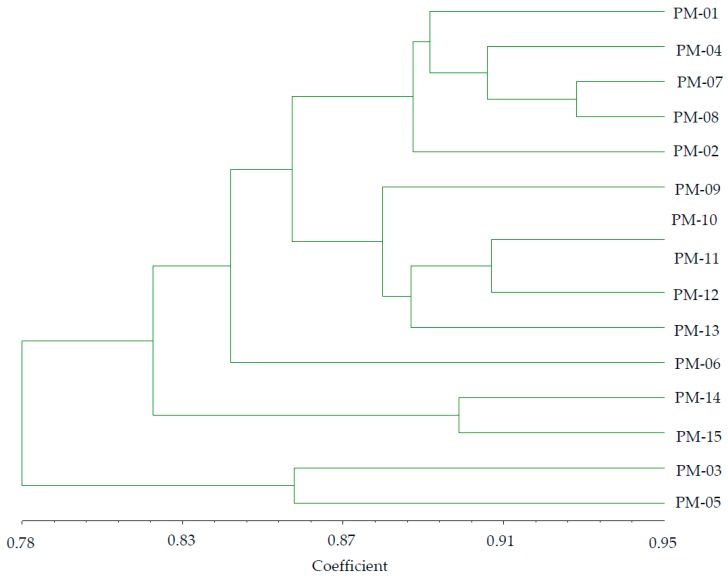
Dendrogram using UPGMA clustering procedures in 15 *P. miliaceum* accessions based on combined ISSR and RAPD markers.

**Figure 6 molecules-24-01012-f006:**
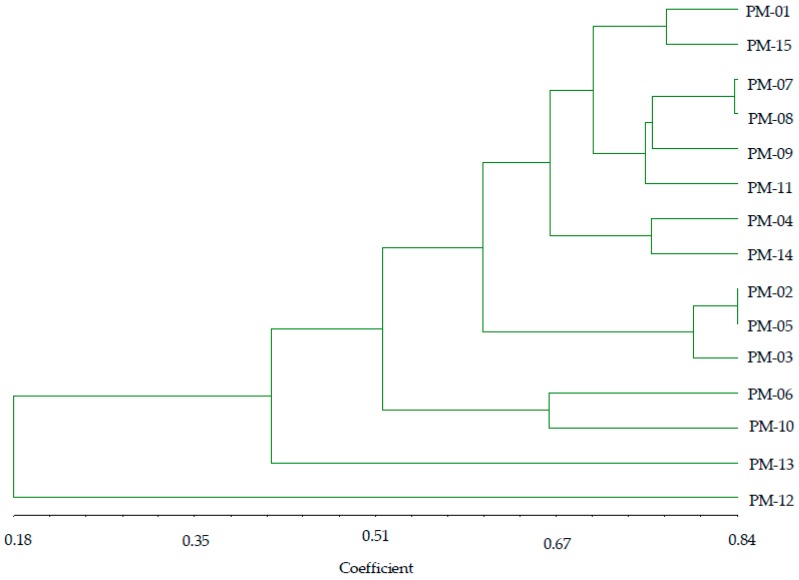
Dendrogram using UPGMA clustering procedures in 15 *P. miliaceum* accessions based on RAPD markers.

**Figure 7 molecules-24-01012-f007:**
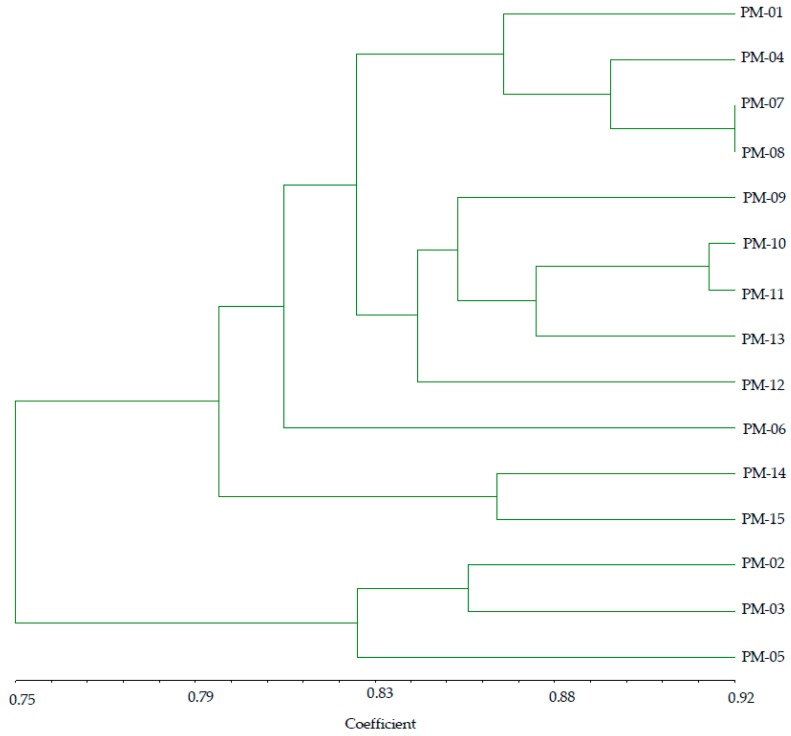
Dendrogram using UPGMA clustering procedures in 15 *P. miliaceum* accessions based on combined ISSR and RAPD markers.

**Table 1 molecules-24-01012-t001:** Morphological characteristics of selected accessions of *P. miliaceum.*

Accession	Bloom Begin (Days)	Full Bloom (Days)	Plant Height (cm)	No. of Leaves	Leaf Length (cm)	Leaf Width (cm)	Ratio (L/B)	Culm Length (cm)	Stem Diameter (cm) **	Spike Length (cm)	Spike Width (cm)
PM-01	85.33±1.53 ^a^	92.33 ± 1.40 ^a^	137.03 ± 2.00 ^h^	6.0 ± 1.0 ^d^	38.3 ± 1.8 ^c^	2.4 ± 0.4 ^e^	15.96	92.00 ± 4.00 ^e,f^	0.8 ± 0.2 ^c^	4.5 ± 0.5 ^h^	45.0 ± 3.0 ^k^
PM-02	85.67 ± 3.06 ^a^	91.67 ± 1.50 ^a^	170.43 ± 2.50 ^l^	8.0 ± 1.0 ^h^	52.2 ± 0.4 ^j^	3.0 ± 0.2 ^f^	17.40	135.50 ± 3.50 ^i^	1.3 ± 0.1 ^h^	4.9 ± 0.1 ^i^	34.8 ± 2.8 ^a^
PM-03	92.67 ± 1.15 ^b^	99.33 ± 1.70 ^b^	115.23 ± 2.10 ^c^	4.5 ± 0.5 ^a^	48.0 ± 1.0 ^h^	3.3 ± 0.5 ^h^	14.55	79.80 ± 2.30 ^c^	0.9 ± 0.2 ^d^	3.7 ± 0.2 ^f^	35.8 ± 3.3 ^b^
PM-04	99.67 ± 2.08 ^c^	105.67 ± 1.53 ^c^	140.30 ± 3.0 ^k^	5.0 ± 1.0 ^b^	51.5 ± 2.5 ^i^	3.3 ± 0.6 ^h^	15.61	96.50 ± 3.50 ^g^	1.1 ± 0.1 ^f^	3.3 ± 0.2 ^c^	44.3 ± 1.8 ^k^
PM-05	98.67 ± 1.53 ^c^	106.33 ± 1.53 ^c^	123.13 ± 2.50 ^d^	6.5 ± 0.5 ^e^	40.8 ± 2.8 ^e^	2.2 ± 0.1 ^d^	18.51	86.80 ± 3.30 ^d^	0.7 ± 0.1 ^b^	3.5 ± 0.5 ^e^	36.5 ± 1.5 ^c^
PM-06	85.67 ± 3.06 ^a^	92.33 ± 1.67 ^a^	135.20 ± 3.0 ^g^	5.5 ± 0.5 ^c^	43.5 ± 1.5 ^f^	3.0 ± 0.0 ^f^	14.50	94.50 ± 4.50 ^f,g^	1.0 ± 0.0 ^e^	3.3 ± 0.1 ^c^	41.0 ± 1.0 ^h^
PM-07	93.33 ± 3.51 ^b^	100.00 ± 1.80 ^b^	186.30 ± 3.50 ^n^	7.5 ± 0.5 ^g^	54.0 ± 2.0 ^l^	3.2 ± 0.2 ^g^	18.00	148.50 ± 1.50 ^j^	1.4 ± 0.1 ^i^	4.5 ± 0.5 ^h^	38.3 ± 1.3 ^e^
PM-08	84.33 ± 2.08 ^a^	92.00 ± 2.00 ^a^	112.33 ± 1.50 ^b^	5.5 ± 0.5 ^c^	31.0 ± 2.0 ^a^	2.10 ± 0.1 ^b c^	14.76	77.00 ± 3.00 ^b^	0.4 ± 0.0 ^a^	3.1 ± 0.3 ^b^	35.5 ± 1.5 ^b^
PM-09	85.33 ± 1.53 ^a^	92.00 ± 1.50 ^a^	111.03 ± 2.00 ^a^	6.5 ± 0.5 ^e^	45.0 ± 3.0 ^g^	3.07 ± 0.1 ^f^	14.66	74.00 ± 2.00 ^a^	1.1 ± 0.1 ^f^	2.6 ± 0.6 ^a^	37.0 ± 3.0 ^d^
PM-10	93.67 ± 2.08 ^b^	99.67 ± 1.00 ^b^	128.30 ± 2.50 ^e^	7.0 ± 0.5 ^f^	54.17 ± 2.0 ^l^	3.09 ± 0.1 ^f^	17.53	90.00 ± 2.00 ^e^	1.41 ± 0.3 ^i^	4.4 ± 0.6 ^h^	38.9 ± 1.1 ^e^
PM-11	93.33 ± 1.53 ^b^	100.00 ± 1.20 ^b^	138.03 ± 1.90 ^i^	7.5 ± 0.5 ^g^	43.5 ± 1.5 ^f^	1.9 ± 0.1 ^a^	22.89	95.80 ± 1.30 ^g^	0.9 ± 0.0 ^d^	4.5 ± 0.5 ^h^	42.3 ± 2.8 ^i^
PM-12	93.33 ± 2.51 ^b^	100.00 ± 2.00 ^b^	182.27 ± 2.50 ^m^	8.5 ± 0.5 ^i^	53.0 ± 1.0 ^k^	3.4 ± 0.1 ^i^	15.59	130.30 ± 1.80 ^h^	1.3 ± 0.2 ^h^	5.0 ± 1.0 ^j^	52.5 ± 7.5 ^m^
PM-13	92.67 ± 2.51 ^b^	100.00 ± 2.00 ^b^	139.00 ± 2.00 ^j^	4.5 ± 0.5 ^a^	39.3 ± 1.8 ^d^	2.17 ± 0.1 ^c,d^	18.11	92.50 ± 4.50 ^e,f^	1.4 ± 0.3 ^i^	3.8 ± 0.8 ^g^	46.5 ± 0.5 ^l^
PM-14	85.00 ± 2.00 ^a^	91.67 ± 1.00 ^a^	115.15 ± 3.00 ^c^	6.5 ± 0.5 ^e^	32.5 ± 2.5 ^b^	2.2 ± 0.4 ^d^	14.77	75.50 ± 1.50 ^a b^	0.8 ± 0.0 ^c^	3.4 ± 0.6 ^d^	39.5 ± 2.5 ^f^
PM-15	85.00 ± 2.00 ^a^	91.67 ± 2.10 ^a^	133.78 ± 1.80 ^f^	6.5 ± 0.5 ^e^	45.5 ± 2.5 ^g^	2.07 ± 0.2 ^b^	21.98	92.50 ± 2.50 ^e,f^	1.2 ± 0.1 ^g^	3.5 ± 0.5 ^e^	40.5 ± 0.5 ^g^

** Experimental data was expressed as mean ± standard deviation (*n* = 3). Data having the same letter in a row were not significantly different as determined by Duncan’s multiple range test (*p* < 0.05).

**Table 2 molecules-24-01012-t002:** Morphological characteristics of the selected accessions of *P. miliaceum.*

Accession	Weight of 1000 Seed (g) ***	Seed Color **	Tiller Number	Leaf Orientation *	Color of Leaves
PM-01	5.43 ± 0.15 ^f^	3	1	1	Green
PM-02	5.95 ± 0.45 ^h^	1	1	3	Green
PM-03	4.41 ± 0.10 ^a^	1	2	2	Light
PM-04	5.24 ± 0.05 ^e^	2	1	1	Green
PM-05	4.92 ± 0.25 ^b^	3	1	2	Green
PM-06	5.43 ± 0.70 ^f^	5	2	1	Green
PM-07	5.75 ± 0.05 ^g^	3	1	2	Green
PM-08	6.36 ± 0.06 ^i^	3	1	2	Light
PM-09	5.07 ± 0.11 ^c,d^	2	1	2	Light
PM-10	5.72 ± 0.11 ^g^	2	1	2	Green
PM-11	5.04 ± 0.05 ^b,c^	2	1	2	Green
PM-12	5.19 ± 0.08 ^d,e^	2	2	2	Light
PM-13	5.75 ± 0.03 ^g^	1	2	2	Dark green
PM-14	5.74 ± 0.06 ^g^	5	2	2	Dark green
PM-15	5.79 ± 0.04 ^g^	5	2	3	Dark green

* 1 = Slightly above the horizontal axis, 2 = almost horizontal with respect to the stem, 3 = slightly below the horizontal axis pointing downwards. ** 1 = Cream, 2 = Yellow, 3 = Orange red, 4 = Brown, 5 = Dark brown. *** Experimental data was expressed as mean ± standard deviation (*n* = 3). Data having the same letter in a row were not significantly different as determined by Duncan’s multiple range test (*p* < 0.05).

**Table 3 molecules-24-01012-t003:** Pearson’s correlation coefficients between the main morphological characteristics in *P. miliaceum* accessions.

Analytes	Bloom Begin **	Full Bloom	Culm Length	Stem Diameter	No. of Leaves	Leaf Length	Leaf Width	Plant Height	Ratio of L/W	Weight of 1000 Seeds
Bloom begin	1	0.995 **	0.190	0.229	−0.041	0.453 *	0.261	0.420 *	−0.092	−0.482
Full bloom		1	0.157	0.172	−0.071	0.383	0.202	0.188	−0.111	−0.477
Culm length			1	0.595 *	0.626 *	0.652 **	0.425	0.980 **	−0.420	0.192
Stem diameter				1	0.315	0.764 **	0.487	0.621 *	−0.031	0.066
No. of leaves					1	0.416	0.122	0.608 *	−0.222	0.142
Leaf length						1	0.763 **	0.630 *	−0.069	−0.249
Leaf width							1	0.404	−0.089	−0.305
Plant height								1	−0.381	0.164
Ratio of L/W									1	−0.288
Weight of 1000 seeds										1

* Correlation is significant at the 0.05 level (2-tailed). ** Correlation is significant at the 0.01 level (2-tailed).

**Table 4 molecules-24-01012-t004:** Average concentration of chlorophyll pigments of the selected accessions of *P. miliaceum* grown in the field.

Accession	Chl a (µg/mL^−1^)	Chl b (µg/mL^−1^)	Ratio of Chl a and Chl b	Carotenoid ** (µg/mL^−1^)
PM-01	412.67 ± 7.02 ^c,d^	20.96 ± 3.00 ^a^	19.49 ± 1.61 ^b,c^	13.49 ± 1.01 ^a^
PM-02	628.15 ± 6.53 ^k^	36.82 ± 2.56 ^f^	16.95 ± 0.41 ^a^	13.26 ± 0.67 ^a^
PM-03	504.44 ± 5.50 ^e^	27.74 ± 2.53 ^b^	18.29 ± 0.95 ^a,b^	13.56 ± 0.96 ^a^
PM-04	607.03 ± 7.00 ^j^	35.60 ± 1.51 ^d,e,f^	17.15 ±0.60 ^a^	12.91 ± 1.83 ^a^
PM-05	594.96 ± 5.56 ^i^	34.28 ± 2.06 ^c,d,e,f^	17.25 ± 0.71 ^a^	12.81 ± 0.20 ^a^
PM-06	521.44 ± 6.50 ^f,g^	29.72 ± 2.06 ^b,c^	17.35 ± 0.70 ^a^	12.44 ± 0.45 ^a^
PM-07	552.16 ± 7.01 ^h^	31.11 ± 3.01 ^b,c,d^	17.45 ± 0.74 ^a,b^	12.86 ± 0.81 ^a^
PM-08	523.79 ± 5.64 ^g^	29.01 ± 3.00 ^b^	18.16 ± 0.71 ^a,b^	12.71 ± 1.14 ^a^
PM-09	625.05 ± 5.00 ^k^	35.96 ± 2.62 ^e,f^	17.66 ± 0.83 ^a,b^	12.64 ± 0.55 ^a^
PM-10	403.44 ± 6.08 ^b,c^	19.61 ± 3.51 ^a^	20.49 ± 1.55 ^c,d^	12.69 ± 0.82 ^a^
PM-11	372.75 ± 7.51 ^a^	17.56 ± 1.50 ^a^	21.61 ± 1.24 ^d^	12.54 ± 0.50 ^a^
PM-12	398.93 ± 9.00 ^b^	19.53 ± 2.50 ^a^	20.31 ± 1.75 ^c,d^	12.21 ± 0.62 ^a^
PM-13	559.17 ± 4.54 ^h^	31.71 ± 1.54 ^b,c,d,e^	17.78 ± 1.16 ^a,b^	12.59 ± 0.53 ^a^
PM-14	422.08 ± 6.54 ^d^	21.55 ± 2.50 ^a^	19.53 ± 1.14 ^b,c^	13.56 ± 0.75 ^a^
PM-15	511.94 ± 8.00 ^e,f^	29.50 ± 3.12 ^b^	17.28 ± 0.75 ^a^	12.45 ± 0.51 ^a^

** Experimental data was expressed as mean ± standard deviation (*n* = 3). Data having the same letter in a row were not significantly different as determined by Duncan’s multiple range test (*p* < 0.05).

**Table 5 molecules-24-01012-t005:** Total phenolic content, total flavonoid content, and antioxidant activity of selected accessions of *P. miliaceum.*

Accession	TPC ^1^ (mg/g)	TFC ^2^ (mg/g)	DPPH (µg/mL)	ABTS (µg/mL) **
PM-01	24.62 ±1.21 ^d^	11.82 ± 1.05 ^c,d,e^	1881.53 ± 19.82 ^h^	888.49 ± 8.05 ^h^
PM-02	31.32±1.15 ^f^	13.28 ± 1.11 ^e,f^	1861.66 ± 7.63 ^g^	716.59 ± 7.66 ^d^
PM-03	46.01 ± 1.00 ^h^	19.58 ± 1.51 ^g^	2304.24 ± 5.17 ^k^	883.63 ± 5.70 ^g,h^
PM-04	15.97 ± 0.95 ^a^	7.19 ± 1.05 ^a^	2490.24 ± 4.64 ^l^	1087.77 ± 9.58 ^j^
PM-05	58.04 ± 1.00 ^e^	52.56 ± 1.50 ^d,e^	206.44 ± 7.72 ^a^	688.08 ± 7.56 ^c^
PM-06	49.81 ± 1.25 ^b^	38.33 ± 1.15 ^a,b^	316.39 ± 13.22 ^b^	673.77 ± 6.01 ^g^
PM-07	16.24 ± 0.86 ^a^	7.22 ± 1.75 ^a^	858.77 ± 6.37 ^c^	822.74 ± 7.51 ^f^
PM-08	23.36 ± 0.60 ^c,d^	10.43 ± 1.78 ^b,c.d^	978.80 ± 9.01 ^d^	719.58 ± 5.05 ^d^
PM-09	30.44 ± 0.67 ^f^	13.08 ± 1.01 ^e,f^	1452. 96 ± 6.00 ^f^	688.19 ± 9.11 ^c^
PM-10	24.30 ± 0.62 ^d^	10.92 ± 1.01 ^c,d,e^	1964.58 ± 9.03 ^i^	772.54 ± 7.50 ^e^
PM-11	23.07 ± 1.00 ^c,d^	10.35 ± 1.52 ^b,c,d^	1967.33 ± 8.08 ^i^	706.46 ± 6.50 ^d^
PM-12	29.43 ± 2.67 ^e,f^	15.02 ± 1.00 ^f^	2245.59 ± 8.50 ^j^	932.55 ± 5.74 ^i^
PM-13	21.61 ± 0.54 ^b,c^	9.61 ± 1.51 ^b,c^	1447.61 ± 11.66 ^f^	682.36 ± 5.85 ^c^
PM-14	36.85 ± 0.97 ^g^	9.79 ± 1.64 ^b,c^	1262.96 ± 7.54 ^e^	624.85 ± 13.14 ^a^
PM-15	31.42 ± 0.61 ^f^	15.40 ± 0.96 ^f^	1436.89 ± 7.00 ^f^	657.57± 7.50 ^b^

** Experimental data was expressed as mean ± standard deviation (*n* = 3). Data having the same letter in a row were not significantly differed by Duncan’s multiple range test (*p* < 0.05). Abbreviations: TPC, total phenolic content; TFC, total flavonoid content; DPPH, 2,2-diphenyl-1-picrylhydrazyl; ABTS. ^1^ TPC is given as gallic acid equivalent (GAE) mg/g of the extracts and fractions, values are the mean ± standard deviation of triplicate tests. ^2^ TFC is given as quercetin equivalent (QE) mg/g of the extracts and fractions, values are the mean ± standard deviation of triplicates.

**Table 6 molecules-24-01012-t006:** Pearson’s correlation coefficients between the antioxidant activity and total phenolic content and total flavonoid content in *P. miliaceum* accessions.

Analytes	TPC ^1^	TFC	DPPH	ABTS	Chl a	Chl b	Ratio of Chl a and b	Carotenoid
TPC	1	0.886 **	0.732 **	0.370	0.122	0.137	−0.221	0.123
TFC		1	0.933 **	0.268	0.223	0.237	−0.299	−0.138
DPPH			1	0.337	0.291	0.299	−0.357	−0.169
ABTS				1	0.064	0.073	−0.092	−0.127
Chl a					1	0.997 **	−0.912 **	0.017
Chl b						1	−0.935 **	0.013
Ratio of Chl a and b							1	−0.06
Carotenoid								1

** Correlation is significant at the 0.01 level (2-tailed). ^1^ Abbreviations: TPC, total phenolic content; TFC, total flavonoid content; DPPH, 2,2-diphenyl-1-picrylhydrazyl; ABTS, 2,2′-azino-bis(3-ethylbenzothiazoline-6-sulfonic acid); Chl a, chlorophyll a; Chl b, chlorophyll b.

**Table 7 molecules-24-01012-t007:** Inter-simple sequence repeat (ISSR) markers used for determining the genetic diversity of *P. miliaceum.*

ISSR Primer	Base Sequence (5′-3′)	Total Amplicon	No. of Polymorphic Amplicon	Percentage of Polymorphism (%)
UBC809	(AGA GAG)^2^AGA GG	15	12	80.00
UBC811	(GAG AGA)^2^GAG AC	6	3	50.00
UBC818	(CAC ACA)^2^CAC AG	8	3	37.50
UBC820	(GTG TGT)^2^GTG TC	3	3	100.00
UBC827	(ACA CAC)^2^ACA CG	10	5	50.00
UBC824	(TCT CTC)^2^TCT CG	4	2	50.00
UBC829	(TGT GTG)^2^TGT GC	4	3	75.00
UBC841	(GAG AGA)^2^GAG AYC	8	7	87.50
UBC844	(CTC TCT)^2^CTC TRC	9	6	66.67
UBC850	(GTG TGT)^2^GTG TYC	6	6	100.00
ISSR2	(GAG)^6^C	11	11	100.00
ISSR3	(GAC)^6^T	9	6	66.67
ISSR4	(GACA)^5^	3	2	66.67
ISSR5	(GTC)^6^A	6	4	66.67
ISSR6	(GTG)^6^C	6	4	66.67
ISSR8	(GTG)^6^A	7	3	42.86
ISSR9	A(CACA)^3^CACTG	4	2	50.00
ISSR10	(GAC)^6^	4	4	100.00
ISSR11	(GACA)^4^	9	6	66.67
ISSR12	(CTC)^7^A	5	4	80.00
ISSR13	(GACA)^4^A	8	2	25.00
ISSR14	(TC)^8^AG	7	6	85.79
ISSR15	TG(TACA)^4^	6	6	100.00
Total		158	110	70.16

**Table 8 molecules-24-01012-t008:** RAPD marker used for genetic diversity of *P. miliaceum.*

ISSR Primer	Base Sequence (5′-3′)	Total Amplicon	No. of Polymorphic Amplicon	Percentage of Polymorphism (%)
RAPD1	CCAGCCGAAC	4	3	75
RAPD2	ATGGATCCGC	4	3	75
RAPD3	BTTGCCAGCC	5	5	100
RAPD4	AGGGAACGAG	4	4	100
RAPD5	AGCGCCATTG	9	8	88.89
RAPD6	CCAAGCTGCC	2	1	50
RAPD7	ACCCGGTCAC	7	6	85.71
RAPD8	GGGCTCATAG	4	3	75
RAPD9	ATGGATCCGC	0	0	0
RAPD10	AGGTGAACGG	3	2	66.67
RAPD11	CGAGTGCCTA	4	4	100
Total		46	39	91.67

**Table 9 molecules-24-01012-t009:** Summary of genetic variation estimated using RAPD and ISSR markers among the *P. miliaceum* accessions.

Molecular Marker	N	Na *	Ne	I	He	uHe
ISSR + RAPD	206	2.00 ± 0.00 ^a^	1.700 ± 0.018 ^c^	0.601 ± 0.007 ^c^	0.411 ± 0.006 ^c^	0.412 ± 0.006 ^c^
RAPD	46	2.00 ± 0.00 ^a^	1.662 ± 0.048 ^a^	0.578 ± 0.290 ^a^	0.391 ± 0.018 ^a^	0.396 ± 0.010 ^a^
ISSR	160	2.00 ± 0.00 ^a.^	1.675 ± 0.015 ^b^	0.581 ± 0.006 ^b^	0.395 ± 0.006 ^b^	0.408 ± 0.006 ^b^

Abbreviations: N = No. of loci, Na = No. of Different Alleles, Ne = No. of Effective Alleles, I = Shannon’s Information Index, He = Expected Heterozygosity, uHe = Unbiased Expected Heterozygosity, ISSR = inter-simple sequence repeat, RAPD = random amplified polymorphic DNA. * Experimental data was expressed as mean ± standard deviation (*n* = 3). Data having the same letter in a row were not significantly differed by Duncan’s multiple range test (*p* < 0.05).

**Table 10 molecules-24-01012-t010:** Location of *P. miliaceum* accessions collected from different provinces of South Korea.

Sl. No.	Accession	Code	Collection Site	Country
1	*P. miliaceum* L.	PM-01	Hwacheon-myeon, Hongcheon-gun	South Korea
2	*P. miliaceum* L.	PM-02	Hoengseong-gun, anheungmyeon saemal	South Korea
3	*P. miliaceum* L.	PM-03	Chuncheonsi dongsanmyeon gunjali	South Korea
4	*P. miliaceum* L.	PM-04	Hoengseong-gun, gabcheonmyeon yuldongli	South Korea
5	*P. miliaceum* L.	PM-05	Kangwondo, hwacheongun gandongmyeon o-eum2li	South Korea
6	*P. miliaceum* L.	PM-06	Kangwondo, injegun, nammyeon bupyeongli	South Korea
7	*P. miliaceum* L.	PM-07	Kangwondo, hongcheongun seoseogmyeon saeng-gog1li	South Korea
8	*P. miliaceum* L.	PM-08	Kangwondo, yang-yang-gun seomyeon hwang-ili	South Korea
9	*P. miliaceum* L.	PM-09	Kangwondo, yang-yang-gun ganghyeonmyeon jeonjin1li	South Korea
10	*P. miliaceum* L.	PM-10	Kangwondo, hongcheongun nammyeon wolcheonli	South Korea
11	*P. miliaceum* L.	PM-11	Kyeongido, yangpyeong-gun cheong-unmyeon bilyong2li	South Korea
12	*P. miliaceum* L.	PM-12	Kangwondo, hwacheongun sangseomyeon mahyeonli	South Korea
13	*P. miliaceum* L.	PM-13	Kangwondo, pyeongchang-gun daehwamyeon haan3li	South Korea
14	*P. miliaceum* L.	PM-14	Kangwondo, pyeongchang-gun daehwamyeon haan5li	South Korea
15	*P. miliaceum* L.	PM-15	Kangwondo, pyeongchang-gun daehwamyeon haan4li	South Korea

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
