# Peer review of "Diversity in Accessions of Panicum miliaceum L. Based on Agro-Morphological, Antioxidative, and Genetic Traits"

_molecules, 2019, doi:10.3390/molecules24061012_

Round 1

Reviewer 1 Report

Manuscript Number: molecules-447551

Title: Diversity in Accessions of Panicum miliaceum L. Based on Agro-morphological, Antioxidative, and Genetic Traits.

General comments:

Very interesting, useful and well written publication.

Authors should give more comments to their results with the chlorophyll content (paragraph 2.2.), antioxidant activity and the rest of the results, as it is known, that it displays pro-health effects. I could not find this. Was the chlorophyll content positively correlated to antioxidant activity? Or not? What is the significance of that? What arises from that data – is it good or not?

In some places, the English demands small corrections, e.g. lines 48-49.

Detailed comments:

Abstract:

Line 14 ‘GAE’ abbreviation is used there for the first time, ant it should be explained in that line. Not in line 19.

Lines 39-40

The articles [9,10,11] cited are not connected with anticancer activity. I could not find any publication strictly in that subject. Make appropriate citation, please.

In some Tables and figures (e.g. Tab. 1, Fig. 1) PM abbreviation for Panicum miliaceum is used, I think that it should be introduced in the text or captions, because the reader must guess.

I am not sure about location of Figure 1: Shouldn’t it be in Materials and Methods? It is mentioned in lines 453 and 458.

Lines 451-454 and 456-460 – the information is doubled, the sentences mean the same. Please connect this.

Author Response

Manuscript ID: molecules-447551

Thank you for providing an opportunity for revising the manuscript in Molecules. We have revised the manuscript carefully and answered/modified/revised/re-write the manuscript as per the suggestions and guidelines of reviewers and editor comments. We shall look forward for your further suggestions. Thank you for your consideration.

General comments:

Very interesting, useful and well written publication.

Authors should give more comments to their results with the chlorophyll content (paragraph 2.2.), antioxidant activity and the rest of the results, as it is known, that it displays pro-health effects. I could not find this. Was the chlorophyll content positively correlated to antioxidant activity? Or not? What is the significance of that? What arises from that data – is it good or not?

Author response

Thank you for the valuable suggestion. As per the reviewer’s suggestion, we have included more comments, added more related citation in chlorophyll content and antioxidant activities of P. miliaceum accessions. Line 471-489 in the text described about the correlation between chlorophyll content and antioxidant properties and its significance to the plant.

COMMENTS FOR THE AUTHOR:

In some places, the English demands small corrections, e.g. lines 48-49.

Author’s response: 

Thank you for the valuable suggestion. We are sorry for inconvenience. Following correction has been made in the manuscript.

Line 48-49: “agronomic value of and taxonomically classifying plant species” is replaced by agronomic value and taxonomically classifying of plant species”

Reviewer 1

Detailed comments:

Line 14 ‘GAE’ abbreviation is used there for the first time, ant it should be explained in that line. Not in line 19.

Author’s response: 

Thank you for suggestions.

Line 14 and 19: Following correction has been made.

mg gallic acid equivalent (GAE)/g of the”

COMMENTS FOR THE AUTHOR:

Lines 39-40

The articles [9,10,11] cited are not connected with anticancer activity. I could not find any publication strictly in that subject. Make appropriate citation, please.

Author’s response: 

We are sorry for inconvenience. Following appropriate citation related to the anticancer activity of P. miliaceum included in the manuscript.

9. Shen, R.; Ma, Y.; Jiang, L.; Dong, J.; Zhu, Y.; Ren, G. Chemical composition, antioxidant, and antiproliferative activities of nine Chinese proso millet varieties. Food and Agricultural Immunology 2018, 29(1), 625–637

10. RAMADOSS, D.P.; SIVALINGAM, N. Vanillin extracted from proso millet and barnyard millet induce apoptosis in HT-29 and MCF-7 cell line through mitochondria mediated pathway. Asian J Pharm Clin Res, 2017, 10(2), 226-229

11. Zhang, L.; Liu, R.; Niu, W. Phytochemical and antiproliferative activity of proso millet. PLOS ONE 2014, 9(8) e104058

COMMENTS FOR THE AUTHOR:

In some Tables and figures (e.g. Tab. 1, Fig. 1) PM abbreviation for Panicum miliaceum is used, I think that it should be introduced in the text or captions, because the reader must guess.

Author’s response:

Thank you for these comments. We are sorry for inconvenience. Detail about the accessions included in the manuscript in the form of table 1 as given below.

Sl.No.

Accession

Code

Collection   site

Country

1

Panicum miliaceum L.

PM-01

Hwacheon-myeon, Hongcheon-gun

South   Korea

2

Panicum miliaceum L.

PM-02

Hoengseong-gun, anheungmyeon saemal

South   Korea

3

Panicum miliaceum L.

PM-03

Chuncheonsi dongsanmyeon gunjali

South   Korea

4

Panicum miliaceum L.

PM-04

Hoengseong-gun, gabcheonmyeon yuldongli

South   Korea

5

Panicum miliaceum L.

PM-05

Kangwondo, hwacheongun gandongmyeon   o-eum2li

South   Korea

6

Panicum miliaceum L.

PM-06

Kangwondo, injegun,  nammyeon bupyeongli

South   Korea

7

Panicum miliaceum L.

PM-07

Kangwondo, hongcheongun seoseogmyeon   saeng-gog1li

South   Korea

8

Panicum miliaceum L.

PM-08

Kangwondo, yang-yang-gun seomyeon hwang-ili

South   Korea

9

Panicum miliaceum L.

PM-09

Kangwondo, yang-yang-gun ganghyeonmyeon   jeonjin1li

South   Korea

10

Panicum miliaceum L.

PM-10

Kangwondo, hongcheongun nammyeon wolcheonli

South   Korea

11

Panicum miliaceum L.

PM-11

Kyeongido, yangpyeong-gun cheong-unmyeon   bilyong2li

South   Korea

12

Panicum miliaceum L.

PM-12

Kangwondo, hwacheongun sangseomyeon   mahyeonli

South   Korea

13

Panicum miliaceum L.

PM-13

Kangwondo, pyeongchang-gun daehwamyeon   haan3li

South   Korea

14

Panicum miliaceum L.

PM-14

Kangwondo, pyeongchang-gun daehwamyeon   haan5li

South   Korea

15

Panicum miliaceum L.

PM-15

Kangwondo, pyeongchang-gun daehwamyeon   haan4li

South   Korea

COMMENTS FOR THE AUTHOR:

I am not sure about location of Figure 1: Shouldn’t it be in Materials and Methods? It is mentioned in lines 453 and 458.

Author’s response:

Line 453 and 458: We are sorry for inconvenience. Figure 1 mentioned in the materials and methods has been removed from the text.

COMMENTS FOR THE AUTHOR:

Lines 451-454 and 456-460 – the information is doubled, the sentences mean the same. Please connect this.

Author’s response:  

Line 451-454 and 456-460: Thank you for important comments. Following duplicated sentences removed from the text.

A total of fifteen accessions of P. miliaceum seeds were collected from the Germplasm Collection Center, Bioherb Research Center, Kangwon National University, South Korea, originating from different geographical landscapes (Fig.1). Different accessions of P. miliaceum were cultivated from 2014-2017, in the experimental farm land of Kangwon National University, South Korea at 20°45′S, 42°51′W”

Reviewer 2 Report

This paper is focused in an interesting study field, the genetic diversity and antioxidant potential of Panicum miliaceum L; however, this paper has some limitations, so, it cannot be accepted for publication in its present form.

 Major comments

1. The main problem with the manuscript stems lack of clarity in the justification and congruence whit the objective. This information must be included in the introduction with clarity and precision in congruence with the objective.

2.      I consider that it is more appropriate to present separately the results and discussion sections.

3.      Authors should rewrite the discussion highlighting their findings

Author Response

This paper is focused in an interesting study field, the genetic diversity and antioxidant potential of Panicum miliaceum L; however, this paper has some limitations, so, it cannot be accepted for publication in its present form.

Major comments

1. The main problem with the manuscript stems lack of clarity in the justification and congruence whit the objective. This information must be included in the introduction with clarity and precision in congruence with the objective.
Author’s response:  

Line 78-96: Thank you for suggestions. The main objectives of the present study and detail about the related information included in the text.

COMMENTS FOR THE AUTHOR:

2.      I consider that it is more appropriate to present separately the results and discussion sections.

Author’s response:  

Line 206-213: Thank you for suggestions. The results and discussion sections separated in the manuscript.

COMMENTS FOR THE AUTHOR:
3.      Authors should rewrite the discussion highlighting their findings

Author’s response:

Thank you for suggestions. Discussion sections of the manuscript modified with highlighting the findings of present research work.

Round 2

Reviewer 2 Report

The authors responded to all comments and suggestions.